# CrPS_4_ Nanoflakes as Stable Direct-Band-Gap 2D Materials for Ultrafast Pulse Laser Applications

**DOI:** 10.3390/nano13061128

**Published:** 2023-03-22

**Authors:** Wenyao Zhang, Yu Zhang, Xudong Leng, Qun Jing, Qiao Wen

**Affiliations:** 1Key Laboratory of Optoelectronic Devices and Systems of Ministry of Education and Guangdong Province, College of Physics and Optoelectronic Engineering, Shenzhen University, Shenzhen 518060, China; 2Xinjiang Key for Laboratory of Solid State Physics and Devices, Xinjiang University, 777 Huarui Street, Urumqi 830017, China

**Keywords:** metal thiophosphates, direct band gap, broadband, ultrafast photonic

## Abstract

Two-dimensional (2D) materials have attracted considerable attention due to their potential for generating ultrafast pulsed lasers. Unfortunately, the poor stability of most layered 2D materials under air exposure leads to increased fabrication costs; this has limited their development for practical applications. In this paper, we describe the successful preparation of a novel, air-stable, and broadband saturable absorber (SA), the metal thiophosphate CrPS_4_, using a simple and cost-effective liquid exfoliation method. The van der Waals crystal structure of CrPS_4_ consists of chains of CrS_6_ units interconnected by phosphorus. In this study, we calculated the electronic band structures of CrPS_4_, revealing a direct band gap. The nonlinear saturable absorption properties, which were investigated using the P-scan technique at 1550 nm, revealed that CrPS_4_-SA had a modulation depth of 12.2% and a saturation intensity of 463 MW/cm^2^. Integration of the CrPS_4_-SA into Yb-doped fiber and Er-doped fiber laser cavities led to mode-locking for the first time, resulting in the shortest pulse durations of 298 ps and 500 fs at 1 and 1.5 µm, respectively. These results indicate that CrPS_4_ has great potential for broadband ultrafast photonic applications and could be developed into an excellent candidate for SA devices, providing new directions in the search for stable SA materials and for their design.

## 1. Introduction

Ultrafast laser pulses can be generated by converting a continuous laser wave into a short pulse train via a mode-locking method that has attracted considerable interest from various fields of science and technology, including material processing, the semiconductor industry, and advanced instrumentation [1,2,3,4,5,6]. A saturable absorber (SA) is a vital component of mode-locking technology. An SA generates ultrafast pulsed lasers through its nonlinear optical properties, which can periodically modulate the circulated light field in the laser cavity and thus satisfy the ever-growing demand for multiple technological applications [7,8,9,10]. As a result of the sustained efforts of scientists to discover SA materials with valuable properties, such as the semiconductor saturable-absorber mirror (SESAM) [11,12], many two-dimensional (2D) layered materials have now been considered as alternative systems [13,14,15,16]. 2D materials are characterized by their chemical diversity and structural complexity, as well as their unique optical and magnetic properties [17,18,19]. Moreover, semiconductor materials with direct band gaps have become important in a range of technologies such as solar cells and lasers. Their strong light absorption and ability to facilitate efficient light emission make them highly desirable for these applications [20,21,22,23,24]. As a result, several direct-band-gap semiconductors have been developed, including black phosphorus (BP) [25,26,27] and transition metal dichalcogenides (TMDs), for designing mode-locked fiber lasers [28] such as WSe_2_ and MoS_2_ [29,30]. BP has gained a great deal of attention on account of its outstanding performance and inherent layer-dependent direct-band-gap energy. However, utilizing BP in practical applications remains a challenge due to its vulnerability to instability when exposed to ambient conditions [31,32,33]. In recent years, to improve the stability of BP, researchers have used various physical processing routes, such as surface passivation, O_2_ plasma etching, ALD, transfer techniques, and self-assembly, to form different capping layers such as graphene, AlO_x_, SiO_2_, TiO_x_, and polymers [34,35]. Scientists have also employed a variety of strategies to chemically protect BP materials, such as covalent functionalization (the formation of P–Ti, P=N, and P–O bonds) and elemental doping, as well as surface treatment [36,37,38]. However, the optimal means of protecting and passivating BP remains to be determined, and there is still an urgent need to discover novel and air-stable 2D materials for use as SAs.

Metal thiophosphates, such as CrPS_4_, are 2D materials with excellent optical, electronic, and magnetic properties [39,40,41,42,43,44]. In the case of monolayer CrPS_4_, the density-functional theory (DFT) describes it as a bipolar ferromagnetic semiconductor with a Curie temperature of 58 K. In other words, CrPS_4_ is a candidate magnetic van der Waals (vdW) material [45]. Moreover, the peculiar in-plane anisotropy makes few-layer CrPS_4_ flakes a birefringent material having monoclinic symmetry with space group *C*2 (No. 5), with two optical axes [46]. In addition, a lack of inversion symmetry in the crystal structure means that CrPS_4_ may be used as a second-harmonic generation optical material. In short, CrPS_4_ has attracted considerable interest in a number of fields due to its superior physical properties. However, to the best of our knowledge, the saturable absorption properties of CrPS_4_ have yet to be fully studied. The potential of ultrafast photonic devices based on CrPS_4_ flakes also remains unexplored. This prompted us to investigate whether CrPS_4_ flakes could be a promising SA material. Furthermore, compared to other SAs with binary elements (WSe_2_ and MoS_2_), the incorporation of a third element in 2D ternary materials might introduce new degrees of freedom, potentially leading to more intriguing device applications.

In the study reported below, theoretical calculations show that CrPS_4_ is a direct-band-gap semiconductor; that is, it can emit almost all energy in the form of light, indicating that it has potentially good saturable absorption properties, like monolayer MoS_2_ and BP. It is also worth noting that, due to their multibonded crystal structure and chemical composition, CrPS_4_ flakes exhibit remarkable stability even when exposed to air; in other words, the material could potentially be air-stable [47]. We then demonstrate the nonlinear optical absorption of CrPS_4_ flakes prepared by the ultrasound-assisted liquid-phase exfoliation (LPE) method via the P-scan technique, and determine the modulation depth and the saturation intensity to be 12.2% and 463 MW/cm^2^, respectively. Finally, by dripping few-layer CrPS_4_ onto D-shaped fiber, a few-layer CrPS_4_-SA is fabricated and applied for passively mode-locked YDFLs and EDFLs, with ultrashort pulse widths of 298 ps and 500 fs, respectively. By such means, we confirm that CrPS_4_ has great potential in broadband ultrafast photonic applications. Our work provides a new paradigm for exploring the applications of metal thiophosphates in mode-locked fiber lasers.

## 2. Results and Discussion

Bulk CrPS_4_ crystallized in the non-centrosymmetric monoclinic space group *C*2 (No. 5) at room temperature. The structure of layered CrPS_4_ is shown in Figure 1a. The crystal structure was composed of distorted CrS_6_ octahedra and PS_4_ tetrahedra connected into a 2D double layer through chemical covalent bonds in a horizontal direction and vdWs forces along the *c* axis. The sheet thickness was 3.69 Å. Structurally, according to previous reports, the weak vdWs gap between layers was about 2.46 Å; this means that the few-layer form could be readily obtained from bulk samples by the LPE method. More importantly, the multibonded crystal structure of CrPS_4_, in contrast to the lone-pair electrons found in BP, gave it the potential for air stability. It is commonly known that layered BP possesses a honeycomb structure in which a phosphorus atom is covalently bonded to three neighboring atoms through their *p*-orbitals, exposing a pair of lone-pair electrons (Figure 1b). The lone pairs of phosphorus atoms can readily react with oxygen to form P_x_O_y_; this ultimately leads to the formation of phosphoric acid and H_2_O, and thereby presents a significant obstacle to the use of BP in applications involving flexible electronics and photoelectronics.

To gain deeper insights into the band-structure properties of CrPS_4_, the CASTEP mode in Material Studio software was utilized to perform DFT calculations of the band structure and density of states (DOS) of both bulk and monolayer forms of CrPS_4_ [48]. The calculated electronic band structures of bulk and monolayer CrPS_4_ are illustrated in Figure 2b,c; these indicate that CrPS_4_ is a direct-band-gap semiconductor, in agreement with previous research [49]. From Figure 2b, it can be seen that the bulk CrPS_4_ nanoflakes exhibited a band-gap energy of 0.97 eV. As the number of layers decreased, the energy of the band gap increased. When the CrPS_4_ nanoflakes were monolayered, the band-gap energy increased to about 2.15 eV. As illustrated in Figure 2d, the DOS plots reveal significant spin splitting in the d-orbitals of Cr atoms, indicating the presence of a sizable spin polarization in the bulk system.

Using the LPE method, few-layer CrPS_4_ nanoflakes were exfoliated simply and effectively. Subsequently, we demonstrated the powder X-ray diffraction (XRD) patterns of CrPS_4_ and its samples after exfoliation treatment by XRD spectroscopy, as illustrated in Figure 3a. It can be seen that the positions of the diffraction peaks on the two patterns were in almost perfect agreement with a standard PDF card of CrPS_4_ (PDF #30-0411), indicating the integrity of the structure after exfoliated CrPS_4_ nanoflakes were obtained. In addition, the diffraction pattern from the CrPS_4_ sample was analyzed using X’pert Highscore Plus 3.0 [50]. The characterization of the diffraction pattern occurred in the range of 2ϴ = 10–70°, as shown in Appendix A. The diffraction pattern from the sample matched well with the diffraction pattern from the previous report. The reference code of CrPS_4_ is 00-033-0404 [51]. The vibrational modes of the CrPS_4_ were verified by the Raman spectrum and detected in the range 200–700 cm^−1^ (excitation wavelength: 532 nm, inVia, Renishaw, Wotton-under-Edge, UK) at room temperature. As shown in Figure 3b, about 13 vibration peaks were found in the Raman spectrum; this was in agreement with previously reported results [46], and further confirmed the rationality of the structure of the CrPS_4_. The surface morphology of the samples as exfoliated was then analyzed via scanning electron microscope (SEM, JSM-5910LV, JEOL, Tokyo, Japan). As depicted in Figure 3c, the CrPS_4_ nanoflakes exhibited an obvious layered structure, indicating that the nanoflakes were successfully fabricated based on the LPE method. The morphology of the CrPS_4_ nanoflakes was also tested using an atomic force microscope (AFM, MFP-3D Infinity, Asylum Research, Oxford, UK), which can observe the lateral size of nanoflakes. Figure 3d,e show CrPS_4_ nanoflakes with an average thickness of ~25 nm. The linear optical transmission spectrum of exfoliated CrPS_4_ was also detected using a UV-vis-NIR spectrophotometer (LAMBDA, Perkin Elmer Inc., Waltham, MA, USA). We found that, at wavelengths of 1030 and 1530 nm, the transmittances were approximately 91.7% and 93.3%, respectively (Figure 3f). BP has attracted tremendous interest because of its natural layer-dependent direct-band-gap energy, but layered BP is unstable and degrades rapidly in ambient conditions within hours. Calculated results indicated that CrPS_4_ is also a direct-band-gap semiconductor, and this made us keen to discover if it was stable in ambient conditions for long time periods. Firstly, taken together, the powder XRD and Raman spectroscopy studies clearly demonstrated that the CrPS_4_ structure remained stable after months in ambient conditions. As shown in Appendix A, the intensity and peak position of CrPS_4_, both as exfoliated and after ~2 months in air, remained essentially unchanged. The intensity and peak position of the Raman modes of CrPS_4_ also remained essentially unchanged for ~2 months, indicating the air-stability of CrPS_4_ (shown in Figure 3b). The CrPS_4_ nanoflakes were then analyzed via energy-dispersive X-ray spectroscopy (EDS, Oxford Instruments, Oxford, UK). As depicted in Appendix A, an average Cr/P/S molar ratio of 1.0:1.0:4.0 was recorded for CrPS_4_ nanoflakes as exfoliated and again after ~2 months in air, further confirming the air-stability of the samples. To more intuitively demonstrate this stability after exposure to air for about 2 months, the surface morphology of the sample was analyzed via SEM. As can be seen in Appendix A, the CrPS_4_ nanoflakes continued to exhibit an obvious layered structure.

The obtained few-layer CrPS_4_ nanoflakes were then dripped onto the D-shaped fiber to form an SA device. To investigate the nonlinear optical properties of the CrPS_4_-SA, a dual-channel balanced detection measurement system based on an erbium-doped fiber laser (1550 nm, 100 fs, 8.05 MHz) was employed, as shown in Figure 4 inset. Equation (1) only considers the case of single-photon absorption, and the nonlinear saturated absorption curve of CrPS_4_-SA was obtained after fitting [52,53].
(1)$$T\left(I\right)=1-\Delta T\times \exp\left(\frac{-I}{I_{sat}}\right)-T_{ns}$$

In Equation (1), *T*(*I*) is the transmission rate, ∆T is the modulation depth (MD), *I* is the input intensity, $I_{sat}$ is the saturated intensity, and $T_{ns}$ is the nonsaturable loss (NL). The fitting results are shown in Figure 4. Values of MD, $I_{sat}$, and NL were found to be ~12.2%, ~463 MW/cm^2^, and ~25.3%, respectively.

Next, to validate the excellent potential of the layered CrPS_4_ for ultrafast laser applications, we constructed 1.0 μm and 1.5 μm all-fiber laser cavities using Er-doped or Yb-doped fibers. The structural diagram of the optical fiber laser structure is shown in Figure 5. The laser cavity consisted of a section of Er-doped or Yb-doped fiber, a laser diode (980 nm Pump Laser, Hanyu, Shanghai, China), a wavelength division multiplexer (WDM, Mingchuang, Shenzhen, China), an optical coupler (OC, Mingchuang, Shenzhen, China), a polarization-independent optical isolator (ISO, Mingchuang, Shenzhen, China), a polarization controller (PC, General Photonics, Losa Angeles, US), and a D-shaped fiber based on CrPS_4-_SA. ISO and PC were used to ensure the unidirectional propagation of light and to adjust its polarization state, respectively. The evanescent field length of the D-shaped fiber optic bare leak was 10 mm, and the distance between the surface and the core was 1 μm. The Er-doped fiber laser cavity length was 21.96 m, including 4 m EDF (Nufern EDFC-980-HP, Hanyu, Shanghai, China) and 17.96 m single-mode fiber (SMF, Hanyu, Shanghai, China), with dispersion parameters at 1530 nm of −12.2 ps/(km·nm) and 18 ps/(km·nm), respectively. The net dispersion was estimated as −0.39 ps^2^. Similarly, the cavity length of Yb-doped laser consisted of 1 m YDF (Nufern SM-YSF-HI-6/125, Hanyu, Shanghai, China) and 15.56 m single-mode fiber (HI1060), and the net dispersion of the cavity was estimated at 0.35 ps^2^. The pulse performance of the laser output was determined by a power meter, a spectrum analyzer (AQ6370C, Yokogawa, Tokyo, Japan), a photodetector (DET08CFC/M 5 GHz, Thorlabs, Newton, MA, USA), a hybrid oscilloscope (DP04104B 1 GHz/s, Tektronix, OR, USA), and autocorrelator monitoring (PulseCheck, APE, Berlin, Germany). The temperature and humidity of the ultra-clean laboratory were 16.8 °C and 58%, respectively.

Importantly, we performed pre-experiments to demonstrate that, in the absence of SA, no mode-locking occurred in the laser cavity, regardless of PC modulation or pump power. By such means, we confirmed the authenticity of the experiment. After SA was applied to the laser cavity, we adjusted the PC while continuously increasing the pump power, and observed the output waveform of the oscilloscope. We achieved continuous-wave mode-locking (CWML) when the pump power was higher than 170 wm. When the pump power was increased to 300 mW, the output power was 10.64 mW, and pulse energy and peak power were 1.174 nJ and 3.94 W, respectively. The mode-locking sequence diagram is presented in Figure 6a, which shows a 110.4 ns time interval between adjacent pulses, which corresponds well to the pulse repetition rate of 9.05 MHz. The inset in Figure 6a shows the amplitude intensity plot of the mode-locked pulse sequence; this indicates that the mode-locked sequence existed stably for a long time. The spectrum with a central wavelength of 1036.1 nm had a 3 dB spectral width of 0.84 nm, as shown in Figure 6b. The autocorrelation trace corresponding to the measured pulse at this time is shown in Figure 6c. It can be seen that the pulse width is about 298 ps, and the time-bandwidth product (TBP) is 69.9, indicating that the pulse has a serious chirp. In Figure 6d, a signal-to-noise ratio (SNR) measurement of approximately 55.3 dB can be observed with a higher signal peak at a laser cavity repetition rate of 9.05 MHz. The relationship between the mode-locked output power and pump power is shown in Figure 6e. When the pump power was 130 mW, the laser cavity output a continuous wave (CW). When the pump power was increased to 170 mW, the output was CWML, and the measured slope efficiency was about 4.9%. Subsequently, we measured the output spectrum of the laser cavity over a longer timescale, at time intervals of 1 h, for a total of 6 h. As shown in Figure 6f, the long-term spectrum was quite stable, indicating that a ytterbium-doped laser has the potential of highly stable operation.

In order to verify that CrPS_4_ could work in a wide wavelength range, we placed the additionally prepared CrPS_4_-SA into an erbium-doped fiber laser cavity for debugging. Stable CWML output was achieved by adjusting the PC when the pump power was above 150 mW. The output power, pulse energy, and peak power were 6.1 mW, 0.893 nJ, and 1786 W, respectively, when the pump power was 270 mW. The output pulse characteristics are shown in Figure 7. The mode-locked pulse sequence is shown in Figure 7a. The pulse period was 146.4 ns, which corresponded to a pulse repetition rate of 6.83 MHz. The inset shows that the mode-locking was quite stable. Figure 7b shows the laser spectrum centered at 1531.6 nm with a 3 dB bandwidth of 5.6 nm. A measured pulse width of 500 fs resulted in a TBP of 0.35, as shown in Figure 7c. It can be clearly observed that the peak value of the high signal at the mode-locked repetition frequency was 6.83 MHz, and the signal-to-noise ratio was about 64 dB, as shown in Figure 7d. Figure 7e shows that the average output power varied linearly with increasing pump power, with a slope efficiency of 2.2%. Subsequently, we also measured the spectrum for 6 h, at time intervals of 1 h, and found that the erbium-doped laser mode-locking was very stable, as shown in Figure 7f.

To demonstrate the stability of CrPS_4_, we placed the previously prepared saturable absorber back into an ytterbium-doped fiber laser, 40 days after the first experiment, and found that the laser could still output mode-locked pulses after adjusting the pump and PC. The mode-locking output results are shown in Appendix A. The threshold of mode-locking was 180 mW, a slight increase (of 10 mW) compared to the first experiment. Appendix A shows the pulse sequence with a pulse interval of 107.2 ns. The inset shows that the laser continued to work with high stability. Appendix A shows that, in comparison with the first experiment, the central wavelength of the mode-locked spectrum remained the same, at 1036.1 nm, while the 3 dB bandwidth changed only slightly, from 0.84 nm to 0.8 nm. The measured mode-locking pulse width was 400 ps, as shown in Appendix A, which was wider than the first measurement. The repetition frequency of the mode-locked strong signal peak was 9.32 MHz, and the SNR was about 52.6 dB, as shown in Appendix A. The relationship between average output power and pump power is shown in Appendix A, and the slope efficiency was 3.6%, which was slightly lower than the first time. We then measured the spectrum for 6 h, and the results are shown in Appendix A. The spectrum remained unchanged, indicating that the laser worked stably, and that the CrPS_4_ saturable absorber we made exhibited high stability.

Once again, 40 days after the first experiment, we put the previously used saturable absorber back into the erbium-doped fiber laser, using the adjustment method described above. To achieve CWML, pump power needed to be increased to more than 150 mw. The mode-locking output characteristics are shown in Appendix A. Appendix A shows that the pulse interval was 136.9 ns, and the inset shows that the laser cavity remained in a stable state. Appendix A shows that the central wavelength of the spectrum was 1531.4 nm, and the 3dB bandwidth was 6.2 nm, which was stable compared with the first experiment. Appendix A shows that the mode-locked pulse width was 594 fs, which was slightly wider than the first experiment, with a TBP of 0.47. The signal-to-noise ratio of a strong signal peak at a repetition frequency of 7.33 MHz was approximately 69 dB. Appendix A shows that the slope efficiency between the average output power and the pump power was 2.4%. Finally, we measured the long-term spectral changes, which indicated the excellent stability of the laser cavity, as shown in Appendix A.

We then compared the data of the two experiments, as set out in Table 1. Although the two experiments were separated by 40 days, we found few differences in the results, indicating that the CrPS_4_-SA we prepared had excellent stability. However, in order to more intuitively observe any variation in the experimental results, we drew a coefficient of variation diagram to represent the degree of dispersion of the pulse parameters of the two experiments, as shown in Figure 8. It can be seen that the coefficients of variation of the pulse width, output power, pulse energy, and coefficients of variation of the Yb-doped fiber laser were close to 30%, while the coefficient of variation of the peak power was as high as 59%. For Er-doped fiber lasers, the coefficients of variation of its pulse parameters were all less than 20%; among these, the coefficients of variation of pulse energy, peak power and slope efficiency were all less than 10%. In short, the output variation of the erbium-doped fiber laser cavity was lower, and this finding is related to the performance of the laser cavity itself. The ytterbium-doped fiber laser cavity worked in the total positive dispersion region, while the erbium-doped laser cavity worked in the anomalous dispersion region, and the optical solitons formed through the balance of dispersion and nonlinear effects were more stable.

The pulse performance of the laser has a crucial influence on the application. In Appendix A, we summarize the performance of mode-locked ytterbium-doped lasers for several representative 2D materials. It can be seen that the output of these ytterbium-doped lasers was in the order of picoseconds. In contrast, the output of our CrPS_4_ ytterbium-doped fiber laser was 298 ps. These results show that CrPS_4_ has higher generation efficiency for ultra-short pulse output, and has certain advantages in terms of pulse width. The maximum output power of our laser was 10.63 mW, which is 28.7 times that of the graphene previously reported by Zhao et al. [54], and 1.38 times that of the Mo_2_C previously reported by Liu et al. [55]. The single-pulse energy of most YDF lasers using 2D materials as SA is usually limited to below 1 nJ; however, our laser achieved an output of 1.174 nJ, with a peak power of 3.94 W, which is 5.5 times that of the WS_2_ reported by Mao et al. [56], and 2.3 times that of the NiPS_3_ reported by Liu et al. [56]. In a similar way, we compared erbium-doped fiber lasers based on other 2D slave materials, as shown in Appendix A. As can be readily observed, because of the remarkable nonlinear optical properties of these 2D materials, their modulation depths vary in percentage terms from low single digits to many tens. In contrast, the modulation depth of our prepared CrPS_4_ was 12.2%, which is higher than the 10.9% for BP reported by Chao et al. [57] and the 5.1% for Mo_2_C reported by Liu et al. [55]. At the same time, our pulse output was 500 fs, which is similar to that achieved by other lasers. In addition, our output reached a level of 6.1 mW, which is twice that of the graphene reported by Bao et al. [7], and 2.3 times that of the BP reported by Chao et al. [57]. The single pulse energy and peak power were 893 pJ and 1786 W, respectively. Not only did we measure the stability of the spectrum over 6 h, we also re-experimented with the previously fabricated saturable absorber 40 days later and found that it still achieved mode locking. These experimental results confirm that CrPS_4_ is a competitive direct-band-gap material with excellent nonlinear optical modulation properties and great potential for broadband ultrafast photonics applications.

## 3. Conclusions

In conclusion, we fabricated high-quality CrPS_4_-SA by the LPE method. Theoretical calculations of the electronic band structures of CrPS_4_ revealed a direct band gap. We studied the applications of few-layer CrPS_4_-SA in ultrafast photonics for the first time. The saturated intensity and modulation depth of CrPS_4_-SA were 463 MW/cm^2^ and 12.2%, respectively, at 1.5 µm. Moreover, based on the excellent saturable absorption of the D-shaped CrPS_4_ SA, the pulse characteristics of fiber lasers operating in conventional soliton states were measured. We successfully obtained picosecond mode-locked pulses of 298 ps and ultrashort femtosecond pulses of 500 fs in the 1 µm and 1.5 µm regions, respectively. The signal-to-noise ratio (SNR) of the mode-locked operation was as high as 55.3 dB at 9.05 MHz (YDFL), and 64 dB at 6.83 MHz (EDFL). More importantly, the few-layer CrPS_4_ exhibited excellent stability during exposure to air for a period of time. Our experimental results show that CrPS_4_ is an air-stable and broadband SA, with promising potential for ultrafast laser applications.

## 4. Experimental Section

**Fabrication of CrPS_4_**. An LPE method was used to exfoliate few-layer CrPS_4_ nanoflakes, in which the vdWs forced between the layers of CrPS_4_ were broken by an ultrasonic wave. Firstly, commercially available high-purity CrPS_4_ (Shenzhen six carbon) powder (about 23 mg) was ground in a mortar and dispersed into *N*-methyl-2-pyrrolidone (NMP, 30 mL), which was exfoliated in an ultrasonic cell disruptor for 20 h at power of 400 W. In order to make the solute to form nanoscale flakes, the solvent was ultrasonicated in an ultrasound cleaner for 24 h. Then, the mixture was centrifugally treated at a speed of 5000 rpm for 20 min to separate precipitation, and the few-layer CrPS_4_ containing supernatant was obtained. All experimental procedures were conducted at room temperature (16.8 °C) and a relative humidity of 58%.

**Characterization**. The powder X-ray diffraction (XRD) patterns of CrPS_4_ and its samples after exfoliated treatment were further demonstrated by XRD spectroscopy; the vibrational modes of the CrPS_4_ were verified by Raman spectrum, and detected in the range 200–700 cm^−1^ (excitation wavelength: 532 nm, inVia, Renishaw, Wotton-under-Edge, UK) at room temperature. The surface morphology of the samples as exfoliated was analyzed via a scanning electron microscope. The morphology of CrPS_4_ nanoflakes was also tested using an atomic force microscope and the linear optical transmission spectrum of exfoliated CrPS_4_ was detected by an UV-vis-NIR spectrophotometer.

**DFT calculation details**. Here, the Vienna Ab initio Simulation Package (VASP, University of Vienna) was utilized to optimize the crystal structures and calculate electronic structures [58,59,60]. The exchange and correlative potentials of electron−electron interactions were accounted for using the generalized gradient approximation (GGA) within the Perdew−Burke−Eruzerhof (PBE) scheme [61,62]. More specifically, an energy cutoff of 500 eV and a Monkhorst–Pack Brillouin zone sampling grid [63] with a resolution 0.02 × 2π Å^−1^ were applied.

## Figures and Tables

**Figure 1 nanomaterials-13-01128-f001:**
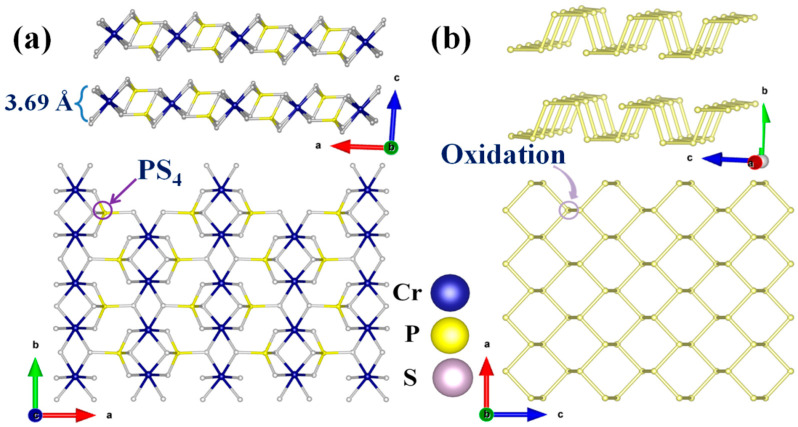
Schematic diagrams of crystal structures of layered (**a**) CrPS_4_ and (**b**) BP. (The arrows represent the three coordinate axes of the crystallographic coordinate system.)

**Figure 2 nanomaterials-13-01128-f002:**
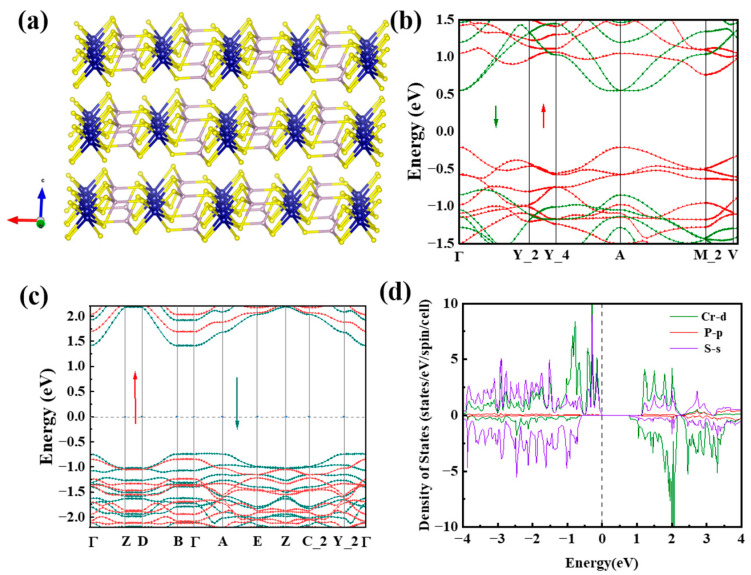
(**a**) Structure of CrPS_4_ (The arrows represent the three coordinate axes of the crystallographic coordinate system); (**b**,**c**) electronic band structures of the bulk and monolayer CrPS_4_ materials calculated by the HSE06 function (The red arrow denotes spin-up bands, while the green arrow represents spin-down bands in the energy band structure.); (**d**) DOS of the bulk CrPS_4_.

**Figure 3 nanomaterials-13-01128-f003:**
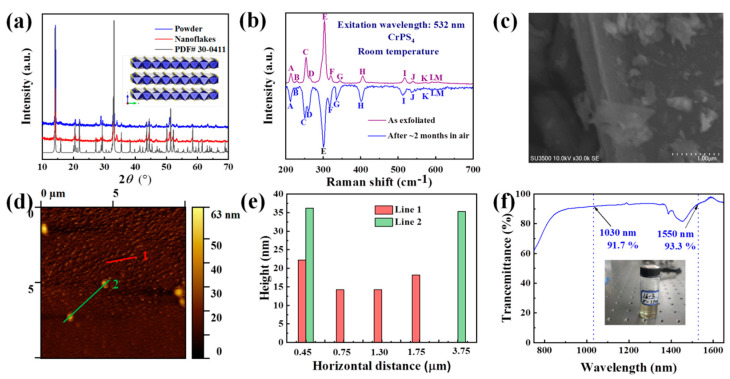
Characterizations of as-synthesized CrPS_4_ nanoflakes after liquid-phase exfoliation: (**a**) the XRD patterns of CrPS_4_; (**b**) the Raman spectrum of few-layer CrPS_4_ nanoflakes on a Si substrate; (**c**) SEM image of CrPS_4_; (**d**,**e**) AFM image of few-layer CrPS_4_ nanoflakes on a Si substrate and corresponding height profile ( The red line and the green line represent the positions of the measured material thickness in the sample); (**f**) UV/Vis/NIR absorption spectrum of few-layer CrPS_4_ nanosheets in NMP.

**Figure 4 nanomaterials-13-01128-f004:**
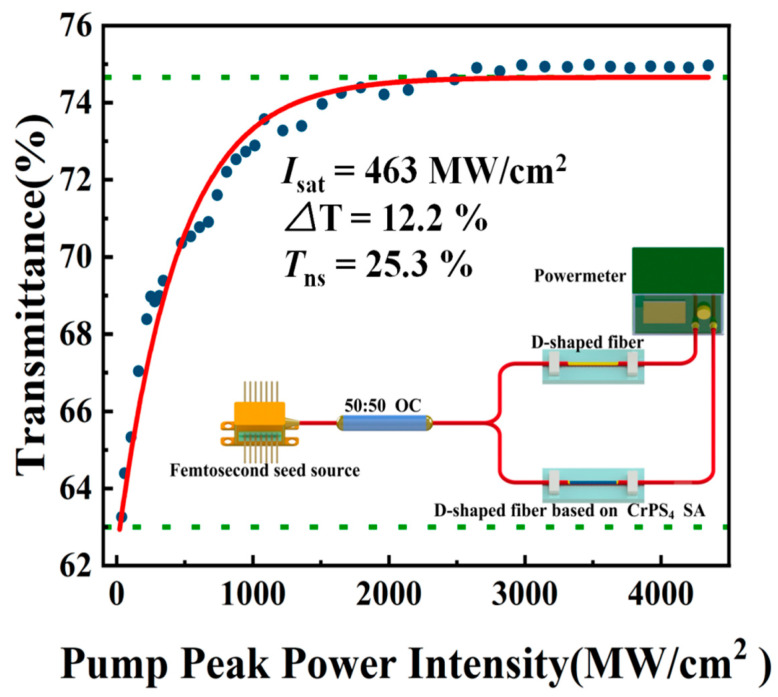
P-scan measurement system diagram and the corresponding nonlinear saturable absorption curve of CrPS_4._

**Figure 5 nanomaterials-13-01128-f005:**
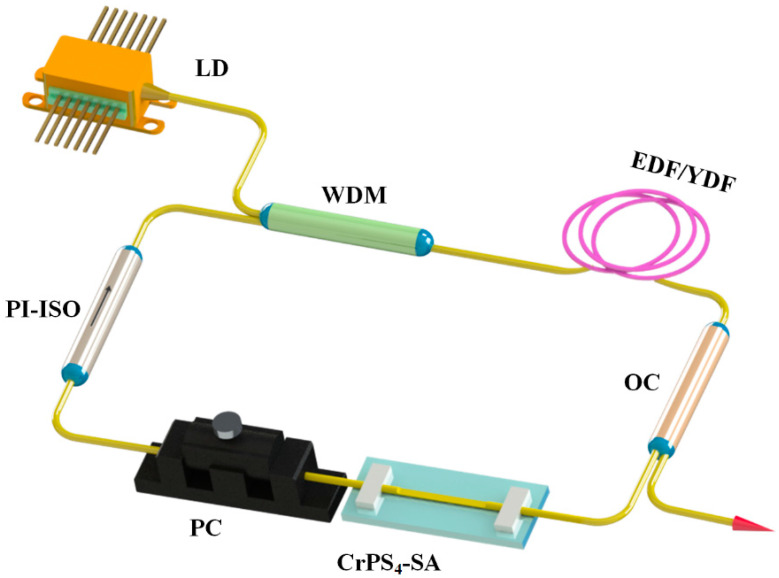
The cavity schematic for the fiber laser.

**Figure 6 nanomaterials-13-01128-f006:**
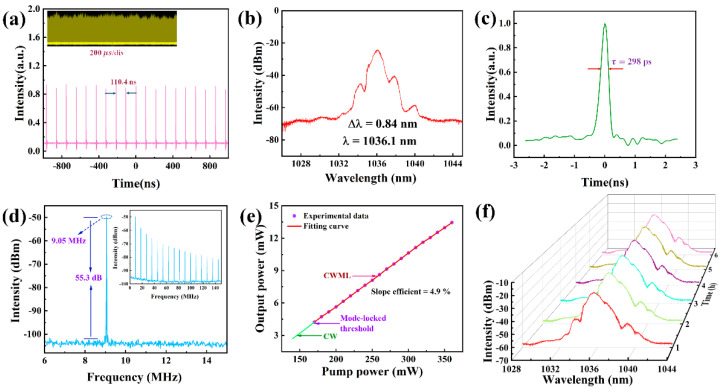
Classical mode-locked output characteristics: (**a**) pulse train; (**b**) optical spectrum; (**c**) pulse width; (**d**) RF spectrum (inset: broadband RF spectrum); (**e**) relative change of output power and pump power; (**f**) long-period spectroscopic measurement (1 h intervals, 6 h in total).

**Figure 7 nanomaterials-13-01128-f007:**
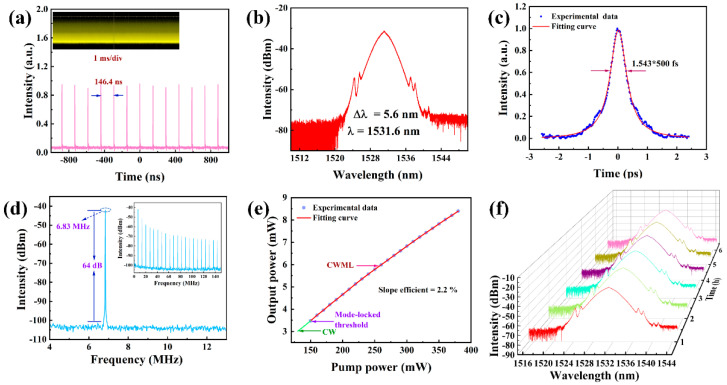
Classical mode-locked output characteristics: (**a**) pulse train; (**b**) optical spectrum; (**c**) pulse width; (**d**) RF spectrum (inset: broadband RF spectrum); (**e**) relative change of output power and pump power; (**f**) long-period spectroscopic measurement (1 h intervals, 6 h in total).

**Figure 8 nanomaterials-13-01128-f008:**
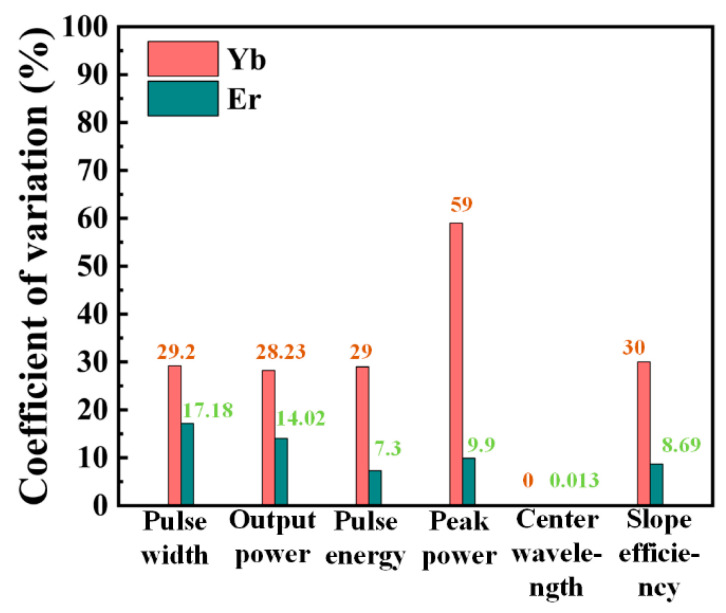
Diagram of the degree of dispersion of the output pulse parameters of the two experiments.

**Table 1 nanomaterials-13-01128-t001:** Comparison of data before and after the two experiments.

Gain Fiber	Experimental Conditions (mW)/Days	Pulse Width	Output Power (mW)	Pulse Energy (nJ)	Peak Power (W)	Center WaveLength (nm)	Slope Efficiency (%)
**Yb**	300/1	298 ps	10.63	1.174	3.94	1036.1	4.9
	300/40	400 ps	8	0.876	2.144	1036.1	3.6
**Er**	270/1	500 fs	6.1	0.893	1786	1531.6	2.2
	270/40	594 fs	7.02	0.961	1617.8	1531.4	2.4

Note: The experimental conditions include the pump power in milliwatts and the time interval between the two experiments.

## Data Availability

The data presented in this study are available on request from the corresponding author.

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
