# Peer review of "CrPS_4_ Nanoflakes as Stable Direct-Band-Gap 2D Materials for Ultrafast Pulse Laser Applications"

_nanomaterials, 2023, doi:10.3390/nano13061128_

Round 1

Reviewer 1 Report

The article titled ‘CrPS4 Nanoflakes as Stable Direct Band Gap 2D Materials for 2 Ultrafast Pulse Laser Applications’ by Zhang et al., demonstrate successful synthesis of air-stable broadband saturable absorber (SA), metal thiophosphates CrPS4, using a liquid exfoliation technique. The manuscript presents judicious synergy between the experimental and simulation works where the authors calculated the electronic band structures and electron density of states to validate the direct band gap. The nonlinear studies were performed sing P-scan technique at 1550 nm and the results gathered are excellent. The experiments are well-planned, and the manuscript is well-structured and presented to the broad audience of nanomaterials. This work presents intriguing novel aspects and hence may be considered for publication. However, the authors must address the below mentioned reviewer comments before this work can be considered for acceptance.

1.     The authors must proof-read the manuscript very carefully. The abstract says ‘The electronic band structures and electron density of states of CrPS4 were calculated were calculated, revealing a direct band gap. Nonlinear absorption coefficient measurements using P-scan technique at 1550 nm..’!

2.     The motivation statement is not clear in the introduction. It is important present the need for this experiment in the context of latest research in this direction.

3.     The authors must clearly present a summary table that compares the performance of the current work with that of the recent ones.

4.     The font size in a particular figure should remain the same to maintain uniformity.

5.     The XRD data presented in Figure 3a needs to be indexed appropriately using Highscore analysis or any other type of analysis with reference codes.

6.     This work presents an interesting case of utility of 2D materials for advanced nonlinear applications, associated references should be given (ACS Appl. Mater. Interfaces 2021, 13, 14, 17046–17061)

7.     There exit different materials with the potential to be investigated for nonlinear application. The authors should present a justifying reason as to why this material was chosen as compared to others in the literature.

8.     The real-time applications of the study should be mentioned with appropriate references to the broad audience of nanomaterials (Nanomaterials 2020, 10(11), 2263)

9.     Figure 3c caption speaks about the Raman spectrum. However, the reviewer is trying to understand if that is a TEM / SEM image. Please provide appropriate d-spacing if it is a TEM image as it is crystalline from XRD.

10.  The details of the experimental procedure should be given, so as to facilitate reproducibility.

11.  The authors must comment on the reproducibility of the work, as flaky structures are used in the measurements.

Reviewer 2 Report

in this manuscript the authors developed and investigated a novel air-stable and broadband saturable absorber  metal thiophosphates CrPS4. The results are okay presented, but are few observations:

-Abstract line 18, typo error 

-many other typo in manuscript should be checked 

-for information presented  starting  with line 75 to line 86, references should be  added as support.

-Fig 2 b-c are hard to be read

-Fig 3 (c) is not a Raman spectrum!

-what is the influence of the ambient temperature variation on the  CrPS4 structure and physical properties ?

-english sentence should be checked in all manuscript, some are too long.

Round 2

Reviewer 1 Report

The authors responded well.